# Ultrafast Growth of Uniform Multi-Layer Graphene Films Directly on Silicon Dioxide Substrates

**DOI:** 10.3390/nano9070964

**Published:** 2019-07-01

**Authors:** Lijie Zhou, Shuai Wei, Chuanyang Ge, Chao Zhao, Bin Guo, Jia Zhang, Jie Zhao

**Affiliations:** 1School of Mechanical and Power Engineering, Harbin University of Science and Technology, No. 52, Xuefu Road, Harbin 150080, China; 2Key Laboratory of Microsystems and Microstructure Manufacturing, Ministry of Education, Harbin Institute of Technology, No. 2 Yikuang Street, Harbin 150080, China; 3College of Engineering, Swansea University, Fabian Way, Swansea SA1 8EN, UK; 4State Key Laboratory of Robotics and System, Harbin Institute of Technology, No. 2 Yikuang Street, Harbin 150080, China

**Keywords:** ultrafast growth rate, uniform graphene, silicon dioxide substrate

## Abstract

To realize the applications of graphene in electronics, a large-scale, high-quality, and uniform graphene film should first be placed on the dielectric substrates. Challenges still remain with respect to the current methods for the synthesis graphene directly on the dielectric substrates via chemical vapor deposition, such as a low growth rate and poor quality. Herein, we present an ultrafast method for direct growth of uniform graphene on a silicon dioxide (SiO_2_/Si) substrate using methanol as the only carbon source. A 1 × 1 cm^2^ SiO_2_/Si substrate square was almost fully covered with graphene within 5 min, resulting in a record growth rate of ~33.6 µm/s. This outcome is attributed to the quick pyrolysis of methanol, with the help of trace copper atoms. The as-grown graphene exhibited a highly uniform thickness, with a sheet resistance of 0.9–1.2 kΩ/sq and a hole mobility of up to 115.4 cm^2^/V·s in air at room temperature. It would be quite suitable for transparent conductive electrodes in electrophoretic displays and may be interesting for related industrial applications.

## 1. Introduction

Graphene is considered to be promising for use in future electronics [1]. To achieve its usage in electrical devices, first, large-scale, high-quality, and uniform graphene film should be placed on the dielectric substrate. Until now, many methods have been developed to grow graphene on metal surfaces and then transfer it onto a dielectric substrate for electrical applications [2,3,4]. However, the transfer process inevitably damages the film by introducing contaminations, wrinkles, and cracks [5]. Recently, graphene was successfully grown on a dielectric surface using the chemical vapor deposition (CVD) method [6]. However, one of biggest barriers is the very long growth duration (e.g., 1–82 h) [6,7,8], which means a low growth rate and limited potential in commercialization. For industrial production, a new method with a fast growth rate that produces an acceptable quality graphene is a must, which means decreased costs and energy consumption, and increased compatibility. Nevertheless, this goal is not easy to achieve, since there is a lack of metallic catalysts on the substrates needed to effectively pyrolyze the carbon source [9,10,11,12]. Liu et al. first reported the direct growth of graphene on a silicon dioxide (SiO_2_/Si) (i.e., 1 × 1 cm^2^) substrate without any metallic catalysts, which took 3 h to be fully covered [9]. In this condition, the carbon precursors were quite difficult to decompose and nucleate on the dielectric substrates. Furthermore, the plasma-enhanced CVD method has been employed to accelerate decomposition of the carbon source (30–60 min), but the resulting graphene was not uniform [13,14]. Many small graphene nanoclusters or nanocrystalline domains have been grown on dielectric substrates [13,14]. Recently, a trace of metal atoms (e.g., copper) were introduced into the reaction, working as catalysts to efficiently accelerate the decomposition of the carbon source [10,11,12]. Meanwhile, these metal atoms were continuously evaporating during the high temperature synthesis process, leading to a lack of residue on the grown graphene and substrates [10,11,12]. Significantly, a fully covered substrate (e.g., 1 × 1 cm^2^) has been achieved in 30–60 min, and the quality of the graphene was comparable with those grown on metal surfaces [4]. This approach provides an additional opportunity for the direct growth of high-quality graphene directly on a dielectric substrate. Nevertheless, the maximum growth rate is ~5.6 µm/s, which is still far behind that of metals (~60 µm/s) [15]. Therefore, there is still no fast method for the direct growth of graphene on dielectric substrates with an acceptable quality. 

The hydrogen (H_2_) gas has been proven to have a crucial role during CVD growth of graphene on metals [16,17]. It has two known roles: one is the etching of the as-grown graphene, which reduces the concentration of the active C species on the catalyst surface and the other is its facilitation of the catalytic decomposition of carbon resources, which provides more active carbon species on the uncovered metal surface [16,17]. Vlassiouk et al. described the dual roles of H_2_ in the growth of graphene on Cu foil, which determined the graphene’s crystallization quality, layer counts, domain morphologies, edge, growth rate, and stack patterns [17]. Moderate H_2_ concentrations favored a fast growth rate and high-quality graphene formations, while a low concentration resulted in no graphene [17]. Jung et al. observed that the growth rate of graphene increased as the partial pressure (equal to concentration) of H_2_ increased during the annealing process [18,19]. However, some researchers have found high-quality graphene with a fast growth rate was only achievable with a decreased H_2_ concentration, and could even occur without H_2_ present during the reaction [20]. As a result, this dilemma has not yet been clearly addressed with respect to the CVD method for the growth of graphene on metals, not to mention with respect to the direct growth of graphene on dielectric substrates. 

Herein, we present an ultrafast method for the direct growth of uniform graphene on the SiO_2_/Si substrate by using methanol as the only carbon source. The role of H_2_ in the direct growth of graphene on the SiO_2_/Si substrate was first investigated. A 1 × 1 cm^2^ SiO_2_/Si substrate square was almost fully covered with graphene within 5 min, resulting in a record growth rate of ~33.6 µm/s. This outcome has been attributed to the quick pyrolysis of methanol, with help of a trace of copper atoms. The as-grown graphene exhibited a uniform thickness with three layers with a sheet resistance of 0.9–1.2 kΩ/sq and a hole mobility of up to 115.4 cm^2^/V·s in air at room temperature. These parameters are higher than those collected from graphene samples directly grown on dielectrics, and also are comparable to those of the CVD-grown graphene on nickel and copper films (Table 1). In addition, the as-grown graphene films would be quite suitable for transparent conductive electrodes in the electrophoretic displays, which require moderate conductivity and mobility. As a result, the proposal method possesses a competitive advantage in related industrial applications.

## 2. Experimental Details

### 2.1. Direct Growth of Graphene on the SiO_2_/Si Substrate 

The growth was carried out in a horizontal tube furnace, in a manner that was the same as our previous paper [21]. A 1 × 1 cm^2^ SiO_2_/Si substrate was ultrasonically cleaned in acetone, isopropanol alcohol, and deionized water for 5 min each, and dried with pure nitrogen gas. Then it was loaded in the center of the horizontal tube furnace, above which a curved piece of Cu foil was suspended, as illustrated in Figure 1. The distance between the copper foil and the SiO_2_ substrate was controlled to be below 100 μm to facilitate enough copper atoms travelling to the substrate after overcoming the gas flow and vacuum forces (Figure 1b) [10]. The tube was first pumped to 150 m Torr and maintained at the same pressure, while 10 sccm H_2_ was applied before annealing. The tube was then heated to 1020 °C at a rate of 15 °C/min. After 20 min of annealing, the H_2_ was stopped. Then 10 sccm Argon (Ar) was used as carrier gas for methanol to introduce the methanol vapor into the tube. The flow rate of the methanol vapor was calculated to be ~1.56 × 10^−6^ mol/min. After a desired duration, the furnace was naturally cooled to room temperature while applying 10 sccm H_2_ again.

### 2.2. Characterization

The as-grown graphene was characterized using optical microscopy (DM4500P, Leica, Germany), transmission electron microscopy (TEM, Tacnai–G2 F30, Philips-FEI Inc., Netherlands, accelerating voltage of 300 kV), Raman spectroscopy (LabRAM XploRA, HORIBA JY, France, incident power of ~1 mW, pumping wavelength of 532 nm), and atomic force microscopy (AFM, Dimension 5000, Bruker, Germany, tapping mode). The species that were used for the Raman, optical, and AFM tests were the as-grown graphene on the original SiO_2_/Si substrates without further annealing, unless addressed in other parts in this paper. The species used for TEM tests were from the transferred graphene on the copper grids.

### 2.3. Fabrication of Field Effect Transistors for Electrical Measurements 

The electrical properties of the graphene were evaluated based on a field effect transistor (FET) configuration. The as-grown graphene film was first patterned into micro-ribbons. Then, Cr/Au metal contacts (10/50 nm) were fabricated onto the micro-ribbons via thermal evaporation to form the bottom-gate FETs. The channel length (*L*) and width (*W*) were measured to be ~27 μm and 70 μm, respectively. To obtain better contact, the devices were thermally annealed at 200 °C in an H_2_/Ar (10/90 sccm) atmosphere for 30 min. The electrical measurements were carried out in air at room temperature using a semiconductor analyser (Keithley 4200–SCS, Tektronix Inc., Beaverton, OR, USA).

## 3. Results and Discussion

Figure 1a illustrates the process of direct growth of graphene on the SiO_2_/Si substrate. In brief, a piece of Cu foil was suspended on the SiO_2_/Si substrate (Figure 1b), which releases copper atoms at high temperatures during the growth process [10]. Methanol vapor was carried into the tube by Ar gas and transported through the tube during growth (arrows in Figure 1a). It worked as the carbon source, which was quickly decomposed with the presence of the catalytic copper atoms. The suspended Cu foil would prevent copper contaminations in both the graphene film and SiO_2_/Si substrates [10]. According to Oshima’s findings, approximately 70% of alcohols would be pyrolyzed within 3 min at temperatures above 1000 °C [22]. This could be further accelerated in the presence of metallic catalysts [23]. The high temperature (i.e., 1020 °C) and presence of the trace of copper atoms in the experiments may have responded to our ultrafast growth process. After the exclusion of stable methane in a previous study, methanol was found to be a good candidate for the carbon source. It can be a source of carbon and hydrogen and act as an inhibitor to amorphous carbon formation [24]. The previous study confirmed that there was no amorphous carbon that was observed in the CVD synthesis process when using methanol as precursor. This is quite different from other hydrocarbons, such as methane [24]. 

The catalytic pyrolysis of methanol is expected to generate a complex mixture, including hydrogen (H_2_), carbon monoxide (CO), carbon dioxide (CO_2_), water and methane [23,24]. According to the thermodynamic equilibrium of the composition at high temperatures (e.g., >750 °C), the content of the main by-product, H_2_, is saturated at ~68.5% [24,25]. The intermediate, CO_2_, will gradually reduce to CO and H_2_O due to the H_2_, leading to the final content of CO reaching up to ~25.5%. The residual CO_2_ and methane were below 5.5% and 0.5% in the composition, respectively, which was reasonably negligible in our experiment. Finally, the reaction of CO + H_2_→C + H_2_O occurred at our set temperature (i.e., 1020 °C) with the copper catalyst, thus leading to highly active deposition of carbon radicals [25]. After the catalytic decomposition in the presence of the evaporated copper, those carbon radicals were nucleated on the SiO_2_/Si substrate at the low energy locations and formed the carbon nuclei and domains as prolonging the duration of the process (Figure 1a). The effective catalytic pyrolysis of methanol generated a large quantity of carbon radicals, leading to the faster growth of graphene in comparison to the traditional methods [9,10,11,12]. In addition, the reduction of the H_2_ in the feedback gas would reduce the etching at the edge of graphene domains, thereby increasing the growth rate further. 

To reveal the effect of H_2_ on this newly developed process, a series of graphene samples were synthesized using different H_2_ concentrations. Figure 2a–d show a series of Raman spectra that were collected from random points on each graphene sample grown using different H_2_ flow rates of 0, 20, 60, and 100 sccm. The peaks at ~1610 cm^−1^ and ~2700 cm^−1^ were assigned to the G band and 2D band, respectively, which confirmed the presence of graphitic carbon [26]. Compared with the exfoliated graphene on the SiO_2_/Si substrate, the shift of G and 2D bands could be attributed to the stress in the graphene plane during growth [24,27]. Another peak at ~1350 cm^−1^ was assigned to the D band, which was activated by the defects via an inter-valley double–resonance process [27]. Generally, the intensity ratio of the D band over the G band (I_D_/I_G_) revealed the degree of the defects and the in-plane crystallite size (L_a_). The parameters that are derived from Raman spectra are accumulated in Figure 2e–g, which can be used to determine the influence of the H_2_ gas. The value of I_D_/I_G_ increased from 0.94 ± 0.30 to 1.79 ± 0.67 (Figure 2e) as the H_2_ flow increased from 0 to 100 sccm, suggesting the degradation of the quality of the graphene [26,27]. This can be attributed to the increase of the etching effect at high H_2_ concentrations, resulting in the formation of structural defects, vacancies, and fragments in-plane and at the edge of as-grown graphene [16,17,28]. The in-plane crystallite size of L_a_ ((L_a_)^2^ = (1.8 ± 0.5) × 10^−9^ × λ_L_^2^(I_D_/I_G_)^−1^) [29] reduced from 12.26 ± 1.74 nm to 8.89 ± 1.26 nm, indicating less crystallinity at a larger H_2_ flow. In addition, the ratio of the 2D band over the G band (I_2D_/I_G_) gradually decreased from 1.74 ± 0.43 to 1.01 ± 0.77 (Figure 2f). Previous results showed that the decreased value of I_2D_/I_G_ may be caused by the increasing layer number and/or degradation of the quality of graphene [25,30]. In our study, all the graphene samples had a constant layer number, which was confirmed by the slightly variable full width at half maximum (FWHM) of the 2D band (i.e., 51–55 cm^−1^, as shown in Figure 2g) [26,30]. Therefore, the decrease of I_2D_/I_G_ could be attributed to the degradation of the graphene. Overall, the Raman results suggested that the presence of H_2_ during the reaction would degenerate the as-grown graphene. According to the decomposition route of methanol, the by-product CO was reduced by the H_2_ to generate the carbon nuclei. This is the main driving force of the graphene growth on the SiO_2_ surface when using methanol as the only carbon source. Additionally, the exclusion of H_2_ in the carrier gas may also have reduced the etching effect and boost the growth rate remarkably. 

Furthermore, we have investigated the morphological evolution of graphene film on the SiO_2_/Si substrate by varying the growth duration. The optical images of graphene films that were obtained by various growth durations (0.5–30 min) are shown in Figure 3. A large quantity of graphene flakes with an average size of 2–6.5 µm appeared in the first 0.5 min (Figure 3a). The enlarged optical image (Figure 3a, inset) shows the approximately hexagonal shape of the graphene domain. Some flakes quickly extended the size to 20–36 µm in the following 2 min (Figure 3b, inset) and merged with adjacent ones within a further 2.5 min (Figure 3c). As the flakes grew in size, a large quantity of new nuclei was deposited on the uncovered area of the SiO_2_ substrate, resulting in the limitation of further growth of the graphene flake. As a result, the numerous nuclei were responsible for our ultrafast growth rate. The substrate was almost fully covered by the graphene film within 5 min (Figure 3c), resulting in an average growth rate of ~33.6 µm/s. To the best of our knowledge, this is the highest growth rate that has ever been reported for the direct growth of uniform graphene film on a SiO_2_/Si surface [9,10,11,12,15]. Further prolonging the growth duration, the graphene film tended to form a uniform film, considering the same reflection in the optical images (Figure 3d–h). Therefore, the thickness (analogous to layer number) of graphene remained unchanged after 5 min growth. This behavior is analogous to the growth of graphene film on Cu foils using alcohols in consideration of a continuous supply of the precursor and copper atoms [31]. 

The systematical Raman measurements revealed the quality, layer number, and defects of graphene samples with different growth durations (Figure 4). The values in Figure 4 are the statistical data (average with error) from five random spectra (points) of each graphene sample. The I_D_/I_G_ increased a little at first and then uniformly decreased down to 0.94 ± 0.11 (Figure 4a), reflecting the constantly improved crystallization of the graphene film as the growth duration increased. This could be attributed to the restoration of the *sp*^2^–hybrid structure in-plane during the high temperature annealing process, which was also used to restore the graphene oxide [32]. Nevertheless, the value of I_D_/I_G_ was still larger than that measured in graphene grown with a long duration or on the metals. This could be attributed to the lack of sufficient metallic catalysts to completely pyrolyze methanol in a short time, resulting in many structural defects in the graphene. The average I_D_/I_G_ of the graphene samples that were grown with 30 min was much smaller than that measured in the reduced graphene oxide and some of the CVD-grown samples [33,34,35]. This indicates that the obtained graphene is more suitable for electronic applications than the reduced graphene oxide and some CVD-grown graphene [33,34,35]. Additionally, the improvement in quality was confirmed by the increase of I_2D_/I_G_ from 0.1 ± 0.002 to 1.86 ± 0.24 (Figure 4b). Simultaneously, these values indicated a multilayer feature of the as–grown graphene, considering that it was usually above 2 in the single-layer graphene [27,30]. In addition, the FWHM(2D) fluctuated slightly at 55.3 ± 1.92 cm^−1^ (Figure 4c), which was similar to the previous results (56.2 ± 1.6 cm^−1^) in three-layer exfoliated graphene [27]. It was almost confirmed to be three layers of as-grown graphene [27]. 

AFM and TEM measurements were performed to determine the layer feature of the graphene films. The thicknesses of the graphene film at the growth durations of 5~30 min were measured to be in the range of 1.5–2.0 nm (Figure 5a shows a thickness of 1.5 nm), which indicates a three-layer feature, considering the deviation in the AFM measurements [36]. Furthermore, the high-resolution TEM image at the back-folded edge (Figure 5b,c) of each graphene sample clearly shows a three-layer feature. A selected area electron diffraction (SAED) pattern has been applied to investigate the crystallographic pattern and orientation of the graphene. A typical SAED image (Figure 5d) shows two sets of hexagonal diffraction patterns, indicating that there is rotational stacking within the region measured [37]. The rotation of the diffraction pattern is determined by many features, such as intrinsic rotational stacking, back-folding of edges, and overlapping domains [37]. Herein, the rotation has been solely attributed to back-folding, considering that there were only two sets of patterns in the SAED images, indicating the single crystal feature of the selected areas [37]. The lattice constant was calculated to be 0.2468 nm from the SAED patterns (Figure 5d), which fit the graphene lattice of 0.247 nm [38].

To evaluate the electrical properties of the as-grown graphene, bottom-gated field effect transistors (FETs) were made. The inset image in Figure 6a shows the FET device’s configuration, in which the graphene works as a channel material. Figure 6a shows a typical transfer property (current of source-drain vs voltage applied on the gate Ids–Vgs) of the FET devices. The Ids decreased uniformly with a positive shifting of the gate voltage and a neutrality point at approximately 30 V. It demonstrated a typical electrical feature of graphene that is measured in air at room temperature. In addition, the corresponding output curve (Ids–Vds, Figure 6b) showed decreasing Ids as the gate voltage increased. Both of characteristics indicated a heavy hole doping (p-type) feature in the graphene. The appearance of a strong p-type doping feature (Figure 6) was possibly due to the adsorption of oxygen and water molecules, since there are a lots of oxygen-containing groups in the graphene plane [7,10]. The sheet resistances measured by the four-terminate devices were found to be in the range of 0.9–1.2 kΩ/sq, which is much smaller than some previous results [39,40,41,42,43,44,45] (Table 1), indicating the promised electrical conductivity. The carrier mobility was calculated according to the equation: *μ* = (*ΔI*_ds_·*L*/*W*)/(*ΔV*_gs_·*V*_ds_·*C*_ox_), where, *C*_ox_ is the silica gate capacitance (1.15 × 10^−8^ F/cm^2^ for a gate oxide thickness of 300 nm). As a result, the hole and electron mobility were approximately 115.4 and 13.7 cm^2^/V·s, respectively, which are higher than those of graphene film grown on some dielectrics (Table 1) [21,41,42,43], and also were comparable to those of the CVD-grown graphene on nickel [46,47] and copper films [48,49,50] (Table 1). In addition, the hole mobility has been improved at least five-fold compared to our previous results within the same growth duration [21]. Specifically, as-grown graphene films could be quite suitable for transparent conductive electrodes in electrophoretic displays, which require moderate conductivity and mobility. Further optimization parameters for growth should be investigated to improve the quality of graphene that is grown over a very short duration. 

## 4. Conclusions

In summary, we have presented an ultrafast method for the direct growth of uniform graphene film on a SiO_2_/Si substrate. The methanol precursor was rapidly catalytically decomposed once it was introduced into the tube, leading to the ultrafast nucleation and growth of the graphene and a record growth rate of ~33.6 µm/s. Meanwhile, the exclusion of H_2_ in the carrier gas reduced the etching of the as-grown graphene domains, thereby improving the crystallization of the graphene. As a result, the trilayer graphene film was of a good quality, with a sheet resistance of 0.9–1.2 kΩ/sq and hole mobility of up to 115.4 cm^2^/V·s in air at room temperature. These graphene films would be quite suitable for transparent conductive electrodes in the electrophoretic displays, which require moderate conductivity and mobility. Therefore, our method possesses a competitive advantage in related industrial applications. 

## Figures and Tables

**Figure 1 nanomaterials-09-00964-f001:**
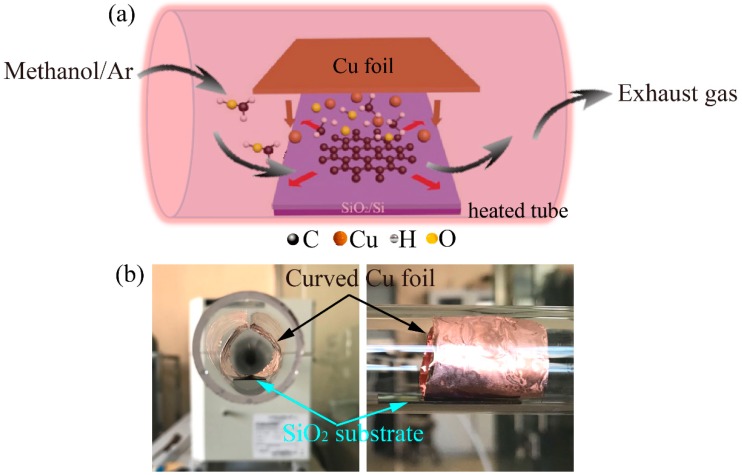
(**a**) Schematic illustration of the possible mechanism of the direct growth of graphene on the SiO_2_/Si substrate using methanol as the precursor. (**b**) Photography of curved suspended copper foil over the SiO_2_ substrate.

**Figure 2 nanomaterials-09-00964-f002:**
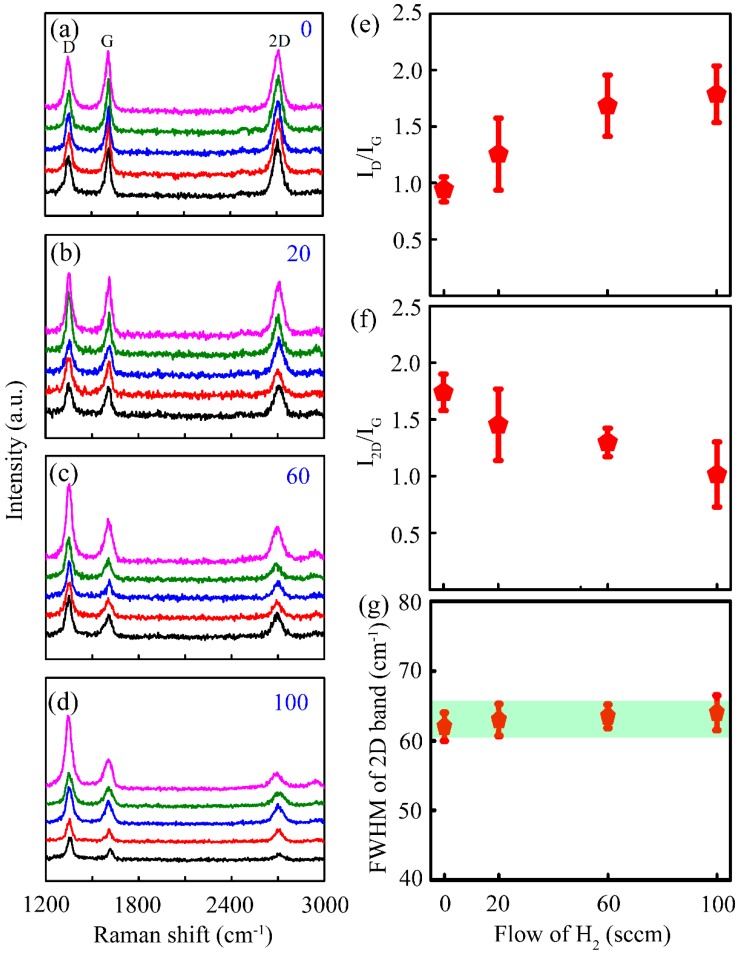
Effect of hydrogen on the properties of the as-grown graphene. Raman spectra of the graphene grown at H_2_ doses of 0, 20, 60, and 100 sccm for (**a**–**d**), respectively. The five different colored curves represent Raman signal collected from five different points on each graphene. The details derived from (**a**–**d**), (**e**) I_D_/I_G_, (**f**) I_2D_/I_G_, and (**g**) and full width at half maximum (FWHM) of the 2D band, respectively.

**Figure 3 nanomaterials-09-00964-f003:**
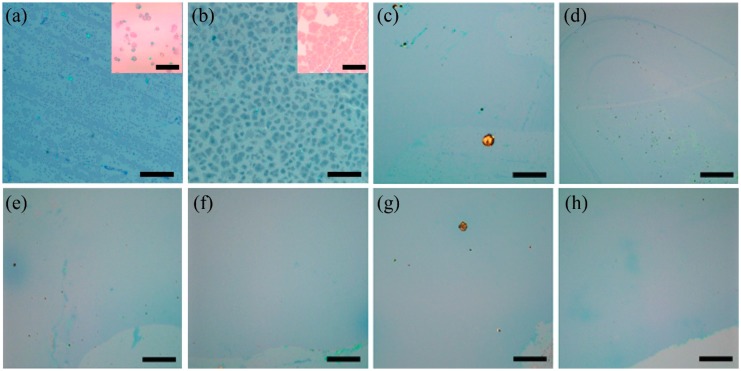
Optical images of the graphene that was directly grown on the 300 nm SiO_2_/Si substrate with respect to the growth durations of (**a**–**h**) 0.5, 2.5, 5, 10, 15, 20, 25, and 30 min. The scale bar in the inset image in (**a**,**b**) are 20 μm, while the others are 200 μm.

**Figure 4 nanomaterials-09-00964-f004:**
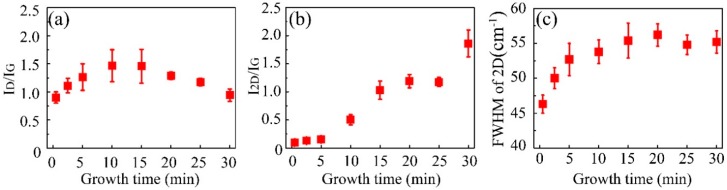
Detailed parameters derived from the Raman spectra of the graphene that was directly grown on the 300 nm SiO_2_/Si substrate (**a**) I_D_/I_G_, (**b**) I_2D_/I_G_, and (**c**) FWHM (2D), respectively.

**Figure 5 nanomaterials-09-00964-f005:**
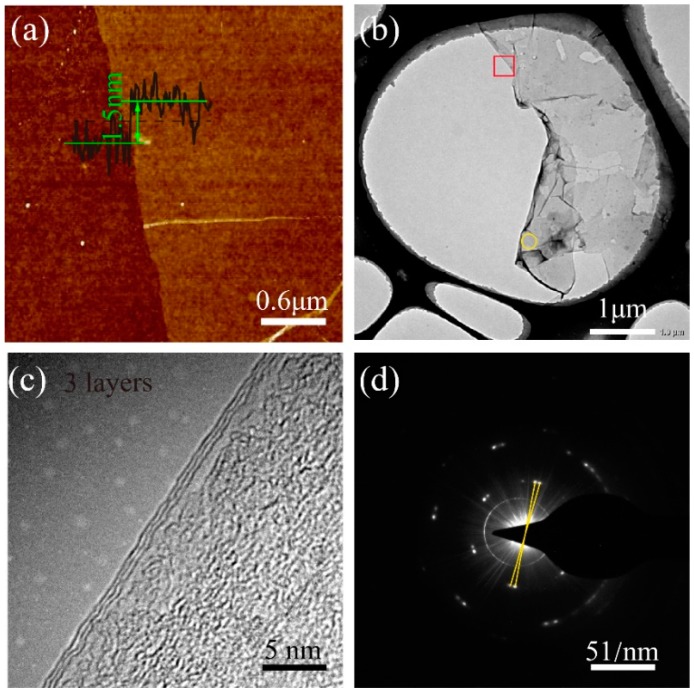
Characterization of the as-grown graphene film that was grown in 30 min. (**a**) A typical atomic force microscopy (AFM) image with the height of the cross-section. (**b**) Low-magnification TEM image. (**c**) High–resolution TEM image at the back-folded edges (red box in b), identifying a trilayer graphene. (**d**) Selected area electron diffraction (SAED) pattern (yellow circle in b) at normal incidence.

**Figure 6 nanomaterials-09-00964-f006:**
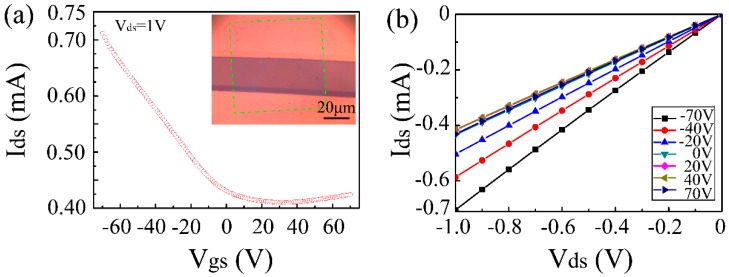
Electrical properties of the as-grown graphene. (**a**) Transfer characteristics (Ids–Vgs) of the field effect transistor (FET) device at Vds = 1 V. Inset is an image of the device configuration with 70 × 70 µm^2^. (**b**) Output characteristics (Ids–Vds) of the graphene device at various gate voltages.

**Table 1 nanomaterials-09-00964-t001:** Comparison of our results with the previous findings.

Substrate	Hole Mobility (cm^2^/V·s)	Sheet Resistance (kΩ/sq)	Ref.
SiO_2_	115.4	0.9–1.2	This work
SiO_2_	1.2–19.1	-	[21]
Al_2_O_3_	16	3.8–6.6	[41]
quartz	3.8	-	[42]
SiO_2_	70	32.7	[43]
SiO_2_	120	2	[44]
SiO_2_	43–580	20–50	[45]
Ni	-	6.6–8.5	[46]
Ni	10–2000	1–2.6	[47]
Cu	-	1.3	[48]
Cu	190	~8.02	[49]
Cu	20–200	-	[50]

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
