# Peer review of "Ultrafast Growth of Uniform Multi-Layer Graphene Films Directly on Silicon Dioxide Substrates"

_nanomaterials, 2019, doi:10.3390/nano9070964_

Round 1

Reviewer 1 Report

The manuscript „Ultrafast Growth of Uniform Graphene Film Direct on Silicon Dioxide Substrates“ by Lijie Zhou at al.  deals with the fast growth of CVD graphene on a SiO2/Si substrate from methanol precursor. In their setup a Cu foil positioned in the close proximity of the substrate delivers Cu atoms that act catalytically and enhance the methanol decomposition on the SiO2/Si surface. The authors claim a record growth rate of 33.6 µm/s. Since fast growth of high quality graphene is a key feature in graphene based technology the manuscript is highly interesting for the involved research community.

However, prior to publication some issues should be solved.

1.     The authors provide the growth rate in units of µm/s, which is quite common in the community. However, the drawback is that this parameter is not clearly defined. In some articles the growth rate refers to the time dependent increase of the lateral domain sizes. Others, refer to their substrate size and the time it takes to form a completely closed layer (irrespective of graphene thickness). Indeed, there are articles where the rates are not comprehensible at all. Therefore, the authors should clearly define how they calculate their growth rate. In the manuscript in the lines 87/88 they state that an 1x1 cm2substrate was almost fully covered after 5 minutes. That would either give a growth rate of  33x103  µm2/s or referring to the length of the substrate a rate of 33.3 µm/s. How do the authors come to the value 33.6 µm/s? This is specifically important since the authors claim a record value, which is very hard to verify under these circumstances.

2.     Is the graphene film after 5 minutes a single layer, multilayer, or a mixed structure?

3.     The authors do not comment on the fate of the Cu foil. Under the given conditions there might also grow graphene/carbon on the Cu foil, which might change its behavior or even completely neutralize its effect.

4.     In the part “2.2. Characterization” the authors mention XPS in line 122. There is no XPS data discussed in the manuscript. Why mentioning XPS then in the experimental part?

5.     Are there lattice constants derived from the SAED patterns? Do they fit to graphene?

6.     Minor things:

Line 48: “resourcesp” should either be “source” or “precursors”  

Line 65: “Therefore, there is still need for a fast method…”

Line 189: “crystallinity” instead of “crystalline”

Please check the manuscript text again for correct English, especially for the correct use of articles and prepositions.

Author Response

Response to Reviewers’ comments

In the following, the reviewers’ comments and suggestions are copied in the font of normal italic, while our response and correction are given in the front of normal bold.

Reviewer 1:

The manuscript „Ultrafast Growth of Uniform Graphene Film Direct on Silicon Dioxide Substrates“ by Lijie Zhou at al.  deals with the fast growth of CVD graphene on a SiO2/Si substrate from methanol precursor. In their setup a Cu foil positioned in the close proximity of the substrate delivers Cu atoms that act catalytically and enhance the methanol decomposition on the SiO2/Si surface. The authors claim a record growth rate of 33.6 µm/s. Since fast growth of high quality graphene is a key feature in graphene based technology the manuscript is highly interesting for the involved research community.

However, prior to publication some issues should be solved.

Comment 1.  The authors provide the growth rate in units of µm/s, which is quite common in the community. However, the drawback is that this parameter is not clearly defined. In some articles the growth rate refers to the time dependent increase of the lateral domain sizes. Others, refer to their substrate size and the time it takes to form a completely closed layer (irrespective of graphene thickness). Indeed, there are articles where the rates are not comprehensible at all. Therefore, the authors should clearly define how they calculate their growth rate. In the manuscript in the lines 87/88 they state that an 1x1 cm2substrate was almost fully covered after 5 minutes. That would either give a growth rate of  33x103  µm2/s or referring to the length of the substrate a rate of 33.3 µm/s. How do the authors come to the value 33.6 µm/s? This is specifically important since the authors claim a record value, which is very hard to verify under these circumstances.

Response: Thanks for the suggestion. As shown in Figure 3a and 3b in the manuscript, the graphene domain directly grown on SiO2/Si substrate shows an approximately hexagonal shape. We defined the length of the diagonal of hexagon as the feature size of the graphene domain. According to the statistical analysis of Figure 3a and 3b, the length of the feature size can be measured and plotted as following (Figure 1). The linear fitting results in a growth rate of 33.6 µm/s, which is at the same level compared with the reviewer’s results. It is noteworthy that the fitting growth rate is an average value, which contains both the extension of the graphene domains and continuous appearance of the new nuclei. 

Figure 1 The feature size of graphene with respect to growth time

Comment 2.     Is the graphene film after 5 minutes a single layer, multilayer, or a mixed structure?

Response: Figure 4b depicts the value of the I2D/IG varied in the range of 0.5-1.25 with the growth time of 5-25 min. This is the normal values for the two or three layers of graphene. In addition, the FWHM(2D) fluctuates slightly at 55.3±1.92 cm–1, indicating a three-layer feature of graphene with growth time of 5-25 min. As a result, the graphene film is multilayer feature after 5 min.

Comment 3.  The authors do not comment on the fate of the Cu foil. Under the given conditions there might also grow graphene/carbon on the Cu foil, which might change its behavior or even completely neutralize its effect.

Response: This is a quite good question. In fact, graphene was also found on the surface of the suspended Cu foil. Nevertheless, we just use the evaporated copper particles as the floating catalyst between the suspended Cu foil and SiO2/Si substrate to accelerate the pyrolysis of methanol. The copper can be evaporated at growth temperature (i.e., 1020 oC) even covered with graphene film.  

Comment 4.   In the part “2.2. Characterization” the authors mention XPS in line 122. There is no XPS data discussed in the manuscript. Why mentioning XPS then in the experimental part?

Response: That is our mistake. At the beginning, we used the XPS to check if there was copper residues on the SiO2/Si substrate after growth process. The XPS data were put in the supporting information in the original manuscript. However, we deleted the supporting information part in the present manuscript but forgot to delete the description in the “experimental details”. We have deleted the description of XPS characterization in the revision.

Correction: In the revision, Line 122-123, “The species those were used for the Raman, optical and AFM tests were the as–grown graphene on the…

Comment 5.     Are there lattice constants derived from the SAED patterns? Do they fit to graphene?

Response: According to the formula, d=Lλ/R, the lattice constant was calculated to be 0.2468 nm from the SAED patterns, which is quite fit to the graphene lattice of 0.247nm [1].

Correction: In the revision, line 281-282,” The lattice constant was calculated to be 0.2468 nm from the SAED patterns (Figure 5d), which is quite fit to the graphene lattice of 0.247nm [38].

Comment 6.     Minor things:

 Line 48: “resourcesp” should either be “source” or “precursors”  

Line 65: “Therefore, there is still need for a fast method…”

Line 189: “crystallinity” instead of “crystalline”

Please check the manuscript text again for correct English, especially for the correct use of articles and prepositions.

Response:

Line 48, The word “resourcesp” should be “source”.

Line 65: The sentence “Therefore, there is still need for a fast method…” was changed into “Therefore, there is still no fast method…. ”

Line189: The word of “crystalline” was replaced by “crystallinity”

Reference

[1] Kevin T. Chan, J. B. Neaton, and Marvin L. Cohen. First-principles study of metal adatom adsorption on graphene. Phys. Rev. B, 2008, 77, 235430.

Reviewer 2 Report

Paper will be of interest for researchers looking at methods to grow graphene. The authors may want to change the title to refelct the material is few or multiple layer graphene. A summary table indicating properties of these films versus those grown with traditional CVD on copper may be of value. 

The use of a growth rate in microns/min is strange and an areal unit would more accurately reflect the growth process.

AFM and XPS experimental data does not appear to have been shown in any figures?

Author Response

Response to Reviewers’ comments

In the following, the reviewers’ comments and suggestions are copied in the font of normal italic, while our response and correction are given in the front of normal bold.

Reviewer 2:

Paper will be of interest for researchers looking at methods to grow graphene. The authors may want to change the title to refelct the material is few or multiple layer graphene. A summary table indicating properties of these films versus those grown with traditional CVD on copper may be of value. 

The use of a growth rate in microns/min is strange and an areal unit would more accurately reflect the growth process.

AFM and XPS experimental data does not appear to have been shown in any figures?

Response: We greatly appreciate you for the constructional comments and suggestions of our manuscript.

According to reviewer’s suggestion, we add the “Multi-Layer” in the title (shown in Line 2 in the revision).

A summary figure which includes several properties of the graphene film is given in Table 1 in the revision.

As shown in Figure 3a and 3b in the manuscript, the graphene domain directly grown on SiO2/Si substrate shows an approximately hexagonal shape. We defined the length of the diagonal of hexagon as the feature size of the graphene domain. According to the statistical analysis of Figure 3a and 3b, the length of the feature size can be measured and plotted as following Figure 1. The linear fitting results in a growth rate of 33.6 µm/s.

Figure 1 The feature size of graphene with respect to growth time

AFM image was shown in Figure 5a in the manuscript, while no XPS data shown in the present manuscript. We have deleted the description of XPS characterization in the “Experimental details”. (Line 122-123)

Reviewer 3 Report

Dear authors,

Despite the "world-record" claim related to the rate of growth of the graphene on the SiO2 layer, not everything is about speed. The quality of the resulting graphene is even of greater importance. In this work, there is little improvement in this regard. The authors state in the manuscript that "the obtained graphene is more suitable for electronic applications due to its higher electrical conductivity"; however the values of mobility are not that great. Actually, the authors wrote (before the conclusions) that further optimization is needed to improve the quality of the graphene resulting from their method. That is the actual key point to be addressed, and the key point missing in this work to contribute with progress to the development of the field, otherwise, this manuscript is just following the steps of existing works (including authors reference 21). Furthermore, the quality of the figures should be improved for a journal like Nanomaterials.

Author Response

Response to Reviewers’ comments

In the following, the reviewers’ comments and suggestions are copied in the font of normal italic, while our response and correction are given in the front of normal bold.

Reviewer 3:

Dear authors,

Despite the "world-record" claim related to the rate of growth of the graphene on the SiO2 layer, not everything is about speed. The quality of the resulting graphene is even of greater importance. In this work, there is little improvement in this regard. The authors state in the manuscript that "the obtained graphene is more suitable for electronic applications due to its higher electrical conductivity"; however the values of mobility are not that great. Actually, the authors wrote (before the conclusions) that further optimization is needed to improve the quality of the graphene resulting from their method. That is the actual key point to be addressed, and the key point missing in this work to contribute with progress to the development of the field, otherwise, this manuscript is just following the steps of existing works (including authors reference 21). Furthermore, the quality of the figures should be improved for a journal like Nanomaterials.

Response: We greatly appreciate you for the constructional comments and suggestions of our manuscript. Up to now, graphene exhibits multiple applications in the electronics, optics, composites, thermal managements, energy storage and conversion. These various applications require different properties of graphene. For example, graphene worked as a filling in the composite will improve the mechanical properties. In the scenario, we highly care about the mechanical properties of the graphene filling and its interface but not for its electrical properties, such conductivity, carriers mobilities. In our experiments, the carriers mobility is about 1-2 orders lower than the record grown on Cu foils, which may not fit for high speed electronics. However, the sheet resistance is 0.9–1.2 kΩ/sq is comparable with single- or bi-layer graphene, which could be worked as the electrodes and applied in monitors and screens. Also, as-grown graphene could be used in thermal transfer and management. Apparently, the quality of graphene does not limit to the carriers mobilities.

In this manuscript, we were focused on developing a fast method for direct growth of graphene on SiO2/Si substrate. The optimization of carbon source and exclusion of H2 in the feedback gas are the new ideals to accelerate growth rate and improve the crystallinity compared with the previous works. This ultrafast growth method may be interesting for the industry.

       In addition, we have improved the quality of figures in the revision.   

Reviewer 4 Report

Re: Nanomaterials - 522233

            Ultrafast growth of uniform graphene film direct on silicon dioxide substrates

            In this paper Zhou at al., present a method for direct growth of uniform graphene on SiO2/Si substrates using methanol as the carbon source and they characterize their samples by Raman spectroscopy, AFM, and TEM. Initially, the effect of H2 was investigated in a series of samples synthesized using different H2 concentrations. Then, the film morphology was studied as function of the growth time.

1.      The work presented here is very similar to reference 21 and lucks innovation.

2.      The growth could be fast, as the authors claim, but at the same time it produces graphene layers with high number of defects. This is evident from the Raman spectra of Figure 2 and Figure 4 (strong presence of the D band in all samples). Therefore, I cannot see how this method could be used for industrial applications where high quality of graphene is a requirement.

3.      Figure 4: It is obvious (Fig.4b) that the growth quality improves as function of time. However, the D-band still is as strong as the G-band (Fig.4a). What is the reason for that? The authors need to discuss more this point.

In summary, even though this work is not innovative and it would be hard to be adopted by the industry, it presents a fast and systematic approach for the growth of graphene on SiO2/Si surfaces. The paper is clearly written and it could be of interest to the graphene community.

Author Response

Response to Reviewers’ comments

In the following, the reviewers’ comments and suggestions are copied in the font of normal italic, while our response and correction are given in the front of normal bold.

Reviewer 4:

Re: Nanomaterials - 522233

Ultrafast growth of uniform graphene film direct on silicon dioxide substrates

In this paper Zhou at al., present a method for direct growth of uniform graphene on SiO2/Si substrates using methanol as the carbon source and they characterize their samples by Raman spectroscopy, AFM, and TEM. Initially, the effect of H2 was investigated in a series of samples synthesized using different H2 concentrations. Then, the film morphology was studied as function of the growth time.

Comment 1.      The work presented here is very similar to reference 21 and lucks innovation.

Response: Thank you for your suggestion and the opportunity for us to state the novelty and differences of our manuscript. To be honest, both our previous paper (Ref. 21) and this manuscript are subject of direct growth of graphene on the SiO2/Si substrate. The previous paper shows that graphene can grow on the SiO2/Si substrate by using ethanol and H2 at atmosphere pressure chemical vapor deposition. The graphene shows a very low carrier mobility of 1.2−19.1 cm2/V·s.

       To improve the quality of graphene, H2 was excluded during the reaction in the present manuscript. It has been proved that H2 has a strong etching effect of as-grown graphene domain during the CVD growth of graphene on metal foils. This etching effect would be response for the low growth rate and bad quality. It was hypothesis that the similar effect of H2 also works in the direct growth of graphene on the dielectrics. In this manuscript, we have detailed the role of H2 during the growth for the first time and verified the hypothesis. As a result, the carrier mobility of as-grown graphene film were increased to 115.4 cm2/V·s in air at room temperature with a sheet resistance of 0.9–1.2 kΩ/sq, which is ~100 times magnitude compared with the previous results and also comparable to some cases in the CVD grown graphene on metals. In addition, to accelerate the decomposition of carbon source, methanol and low pressure condition were taken, resulting in an ever fast growth rate of ~33.6 µm/s, while does not degrade the quality much.

       Above all, the findings in the present manuscript is quite promising compared to the previous paper. Therefore, we think there is update of our growth method.

Comment 2.  The growth could be fast, as the authors claim, but at the same time it produces graphene layers with high number of defects. This is evident from the Raman spectra of Figure 2 and Figure 4 (strong presence of the D band in all samples). Therefore, I cannot see how this method could be used for industrial applications where high quality of graphene is a requirement.

Response: Up to now, graphene exhibits multiple applications in the electronics, optics, composites, thermal managements, energy storage and conversion. These various applications require different properties of graphene. For example, graphene worked as a filling in the composite will improve the mechanical properties. In the scenario, we highly care about the mechanical properties of the graphene filling and its interface but not for its electrical properties, such conductivity, carriers mobilities. In our experiments, the carriers mobility is about 1-2 orders lower than the record grown on Cu foils, which may not fit for high speed electronics. However, the sheet resistance is 0.9–1.2 kΩ/sq is comparable with single- or bi-layer graphene, which could be worked as the electrodes and applied in monitors and screens. Also, as-grown graphene could be used in thermal transfer and management. Apparently, the quality of graphene does not limit to the carriers mobilities.

In this manuscript, we were focused on developing a fast method for direct growth of graphene on SiO2/Si substrate. The optimization of carbon source and exclusion of H2 in the feedback gas are the new ideals to accelerate growth rate and improve the crystallinity compared with the previous works. This ultrafast growth method may be interesting for the industry.

Comment 3. Figure 4: It is obvious (Fig.4b) that the growth quality improves as function of time. However, the D-band still is as strong as the G-band (Fig.4a). What is the reason for that? The authors need to discuss more this point.

Response: As shown in Figure 4a, the value of ID/IG decreases gradually from 1.46±0.31 to 0.94±0.11 as prolonging the growth duration, which is also an index of improvement of crystallinity of graphene. Nevertheless, the final value is still larger than that measured in graphene grown on metals. It would be attributed to the lack of sufficient metallic catalysts to completely pyrolyze methanol in a short time, resulting in many structural defects (strong D band) in graphene.

Round 2

Reviewer 3 Report

This reviewer also agrees with the authors that graphene sheets are required in other fields like composites, thermal management, energy storage, and conversion... but this is not the target of the paper according to the introduction and abstract: if the paper is started (abstract and introduction) highlighting the importance of graphene in electronics, but the conductivity values that are achieved with this method are not paired with the actual needs, the reader may be quite confused. As I mentioned previously, the speed of growth cannot be traded for many other properties demanded by the industry, otherwise, this work would be a mere experimental exercise. Please improve the focusing of the manuscript on its target application/applications and probe the achievements with regard to that application:

In general, less than 500Ohm/Sq is needed for flexible displays. We may admit that there are certain applications in electronics where this requirement can be relaxed up to 1kOhm/Sq.  If this is the target case,  it will be ideal to show what those applications are (electrophoretic displays for example), and how the resistivity of this work compares to those industry requirements. In this manner, you can provide additional value and differentiate more with regard to your previous works. Are you targeting low-refresh-rate flexible displays? Then please show (or at least comment on) the need of stability of the resistivity of a macroscopic sample of your graphene with bending cycles; Are you targeting solar applications? Then please show (or at least comment on) data on the transmission; are you targeting other applications? Then please show the relevant magnitudes required in those fields... As you pointed well in your introduction the field of synthesis of graphene, even on SIO2 substrates, is already nourished with many works and it is important to provide additional value with respect to what is already published (especially for a Q1 journal). This reviewer appreciates new Table1 highlighting the fact that the achievements, in terms of sheet resistance, outperform the existing results considering direct growth on insulator. I recommend to complete it with other fabrication approaches including transfer-based approaches as a reference (highlighting the fact you need a substrate transfer step).

The quality of Figure 6 is not at the level that one may expect for MDPI Nanomaterials, please improve it. (definition, size)

Author Response

Response to Reviewers’ comments

In the following, the reviewers’ comments and suggestions are copied in the font of normal italic, while our response and correction are given in the front of normal bold.

Comment 1: This reviewer also agrees with the authors that graphene sheets are required in other fields like composites, thermal management, energy storage, and conversion... but this is not the target of the paper according to the introduction and abstract: if the paper is started (abstract and introduction) highlighting the importance of graphene in electronics, but the conductivity values that are achieved with this method are not paired with the actual needs, the reader may be quite confused. As I mentioned previously, the speed of growth cannot be traded for many other properties demanded by the industry, otherwise, this work would be a mere experimental exercise. Please improve the focusing of the manuscript on its target application/applications and probe the achievements with regard to that application:

In general, less than 500 Ohm/Sq is needed for flexible displays. We may admit that there are certain applications in electronics where this requirement can be relaxed up to 1kOhm/Sq.  If this is the target case,  it will be ideal to show what those applications are (electrophoretic displays for example), and how the resistivity of this work compares to those industry requirements. In this manner, you can provide additional value and differentiate more with regard to your previous works. Are you targeting low-refresh-rate flexible displays? Then please show (or at least comment on) the need of stability of the resistivity of a macroscopic sample of your graphene with bending cycles; Are you targeting solar applications? Then please show (or at least comment on) data on the transmission; are you targeting other applications? Then please show the relevant magnitudes required in those fields... As you pointed well in your introduction the field of synthesis of graphene, even on SIO2 substrates, is already nourished with many works and it is important to provide additional value with respect to what is already published (especially for a Q1 journal). This reviewer appreciates new Table1 highlighting the fact that the achievements, in terms of sheet resistance, outperform the existing results considering direct growth on insulator. I recommend to complete it with other fabrication approaches including transfer-based approaches as a reference (highlighting the fact you need a substrate transfer step).

Responses: We highly appreciate the reviewer’s comments and suggestions. For our direct growth of graphene film, the reviewer summarized several potential applications. Among those applications, our graphene film may be applied in electrophoretic displays. Recently, the electrophoretic displays are widely used in e-book, which has a requirement of moderate conductivity around 1 kΩ/sq and a low mobility. In this manuscript, we illustrated the graphene film can be grown on SiO2 substrate with a record growth rate. Analogously, the proposal method can be employed using other dielectric substrates, such as quartz, glass, SiC. Therefore, those transparency substrates with a nanometer-thick conductive film are very applicable to the transparent conductive electrodes in electrophoretic displays. Accordingly, we have revised the manuscript by strength a specific application of the transparent conductive electrodes in electrophoretic displays. In addition, we have added the comparison of sheet resistance and mobility of the CVD-grown graphene on the nickel and copper film in Table 1 in the revision.

Correction: In the revision,

line 27-30, “…cm2/V·s in air at room temperature. It would be quite suitable for the transparent conductive electrodes in electrophoretic displays and may be interesting for the related”.

Line 93-100, “temperature. These parameters are higher than those collected from graphene samples direct grown dielectrics, and also are comparable to those of the CVD–grown graphene on nickel and copper films (Table 1). In addition, as-grown graphene films would be quite suitable for the transparent conductive electrodes in the electrophoretic displays, which requires a moderate conductivity and mobility. As a result, the proposal method possesses a competitive advantage in the related industrial application.

Line 316-323, “approximately 115.4 and 13.7 cm2/V·s, respectively, which are higher than those graphene film grown on dielectrics (Table 1)[21, 41, 43-45], and also are comparable to those of the CVD–grown graphene on nickel [46, 47] and copper films [48-51] (Table 1). In addition, the hole mobility is improved by at least 5 times compared with our previous results within the same growth duration [21]. Specifically, as-grown graphene films would be quite suitable for the transparent conductive electrodes in the electrophoretic display, which requires a moderate conductivity and mobility. Further optimization parameters for the growth should be investigated

Line 342-346, “and hole mobility of up to 115.4 cm2/V·s in air at room temperature. These graphene films would be quite suitable for the transparent conductive electrodes in the electrophoretic display, which requires a moderate conductivity and mobility. Therefore, our method possesses a competitive advantage in the related industrial application.”

Comment 2: The quality of Figure 6 is not at the level that one may expect for MDPI Nanomaterials, please improve it. (definition, size)

Responses: The Figure 6 was replotted and exported with 600 dpi resolution in the revision. 

Correction: In the revision, line 326. 

Round 3

Reviewer 3 Report

At this point I consider the paper ready for publication. No further comments.